# Bioelimination of Phytotoxic Hydrocarbons by Biostimulation and Phytoremediation of Soil Polluted by Waste Motor Oil

**DOI:** 10.3390/plants12051053

**Published:** 2023-02-27

**Authors:** Gladys Juárez-Cisneros, Blanca Celeste Saucedo-Martínez, Juan Manuel Sánchez-Yáñez

**Affiliations:** Environmental Microbiology Laboratory, Chemical-Biological Research Institute, Ed-B-1, Michoacán University of San Nicolás de Hidalgo, Morelia C.P. 58060, Michoacán, Mexico

**Keywords:** soil, aliphatic/aromatic hydrocarbons, cometabolism, mineralization, phytodegradation

## Abstract

Soils contaminated by waste motor oil (WMO) affect their fertility, so it is necessary to recover them by means of an efficient and safe bioremediation technique for agricultural production. The objectives were: (a) to biostimulate the soil impacted by WMO by applying crude fungal extract (CFE) and *Cicer arietinum* as a green manure (GM), and (b) phytoremediation using *Sorghum vulgare* with *Rhizophagus irregularis* and/or *Rhizobium etli* to reduce the WMO below the maximum value according to NOM-138 SEMARNAT/SS or the naturally detected one. Soil impacted by WMO was biostimulated with CFE and GM and then phytoremediated by *S. vulgare* with *R. irregularis* and *R. etli*. The initial and final concentrations of WMO were analyzed. The phenology of *S. vulgare* and colonization of *S. vulgaris* roots by *R. irregularis* were measured. The results were statistically analyzed by ANOVA/Tukey’s HSD test. The WMO in soil that was biostimulated with CFE and GM, after 60 days, was reduced from 34,500 to 2066 ppm, and the mineralization of hydrocarbons from 12 to 27 carbons was detected. Subsequently, phytoremediation with *S. vulgare* and *R. irregularis* reduced the WMO to 86.9 ppm after 120 days, which is a concentration that guarantees the restoration of soil fertility for safe agricultural production for human and animal consumption.

## 1. Introduction

Soil contamination by petroleum hydrocarbon (PHC) mixtures such as waste motor oil (WMO) is a serious environmental and economic problem worldwide [1]. WMO is a hazardous waste composed of a heterogeneous mixture of linear, branched, aromatic, and polycyclic aliphatic PHCs, which are generally toxic to the environment as well as to all forms of life, including humans [2,3,4]. The properties of soil polluted by WMO are affected, as WMO clogs pores, which prevents aeration, the diffusion of liquids, and the mineralization of organic matter and causes a loss of fertility [3,5].

Currently, the remediation of soil contaminated by WMO represents a global challenge due to the risk to human health, specifically regarding the ease of absorption by dermal contact and the mutagenic and/or carcinogenic effects of some of its aromatic PHCs. Therefore, it is necessary to find an ecologically sustainable remediation strategy for mineralizing all types of PHCs in WMO [4,6] in accordance with environmental regulations. In Mexico, the NOM 138-SEMARNAT/SS regulation [7] establishes that the maximum permissible limit of PHC in soil is 4400 ppm.

To solve this environmental problem, there are physical remediation methods such as solvent extraction, air spraying, thermal desorption, microwave heating, and chemical methods such as Fenton’s reagent, ozone, and photocatalytic degradation, which, although effective, are expensive and generate secondary pollutants [8]. In contrast, biological techniques such as bioelimination, better known as bioremediation, are environmentally safe for the mineralization of the WMO hydrocarbons in soil into harmless compounds. The above depends on the survival of native microorganisms capable of mineralizing a diversity of PHCs, as well as associations with plants for the phytoremediation of soil contaminated by different concentrations of PHCs [9]; both of these are useful and allow for the exploitation of the natural potential of microorganisms and plants for the mineralization of WMO [10].

The ecological techniques for soil biorecovery are biostimulation and phytoremediation. Applied individually, they are usually not enough to solve the problem of restoring the productivity of agricultural soil. Therefore, the use of two or more methods in a proper interaction guarantees the efficiency of the biorecovery of soil polluted by WMO [3]. Therefore, biostimulation enriches the soil with basic minerals such as N (nitrogen), P (phosphorus), and K (potassium) to induce heterotrophic aerobic microorganisms native to the soil to mineralize the WMO. Another form of biostimulation is the application of a fungal crude extract (CFE) containing the extracellular enzymes of mitosporic ligninolytic fungi (LMF), from *Aspergillus* sp. [11], *Fusarium solani*, and *Penicillum chrysogenum*, which generate: (a) manganese peroxidase (MnP), which hydrolyzes phenol by the oxidation of Mn^2+^ to Mn^3^; (b) lignin peroxidase (LiP) and (c) laccase (p-diphenol-dioxygen: oxide reductase), which hydrolyze WMO aromatics [12], including polycyclic aromatics [13,14]; and (d) green manure (GM), which balances the C:N ratio of the soil to facilitate the mineralization of the WMO in adverse soil environmental conditions [15]. This is followed by phytoremediation using a plant to mineralize the WMO remaining after biostimulation. The selected plant, such as *Sorghum vulgare*, must tolerate high concentrations of PHC through deep and fibrous roots and specific physiological and genetic properties of gramine [16]. In addition, the phytoremediation of soil polluted by WMO using *S. vulgare* could be enhanced with microorganisms that promote plant growth, such as *Rhizobium etli*, that synthesizes phytohormones within the roots while at the same time mineralizing some aliphatic PHCs, including aromatics, by cometabolism [17,18]. The phytoremediation of WMO-contaminated soil by *S. vulgare* could be enhanced with *Rhizophagus irregularis*, an endomycorrhizal fungus capable of solubilizing PO_4_^−3^ in addition to being tolerant to the phytotoxicity of WMO aliphatic and aromatic hydrocarbons [19,20]. These microorganisms support the tolerance of plants to PHCs, improving mineral uptake and helping the plants to cope with oxidative stress due to the PHCs of WMO [21] and thus achieving effective soil biorecovery.

Based on these facts, the objectives of this work were to achieve (a) the biostimulation of soils contaminated by 34,500, 65,418, and 89,830 ppm of WMO by applying a crude fungal extract to hydrolyze WMO aromatics, followed by the biostimulation with *Cicer arietinum* as a green manure, and finally, (b) phytoremediation using *Sorghum vulgare* with *Rhizophagus irregularis* and/or *Rhizobium etli* to reduce the remaining 34,500 ppm WMO concentration to one equal to the value of NOM-138 SEMARNAT/SS [7] or to reduce the value to lower than that naturally detected, allowing the reuse of the soil in the production of safe human or animal food.

## 2. Results

Figure 1A presents the remaining WMO concentrations in soil impacted by (A) 89,533, (B) 65,418, or (C) 34,500 ppm of WMO after biostimulation by CFE and GM for 60 days. When agricultural soil polluted by 34,500 ppm of WMO was biostimulated with both *C. arietinum* applied as a GM and CFE, the concentration decreased to 2066 ppm, a numerical value with a statistical difference when compared to soil biostimulated only with CFE, for which the remaining WMO was 10,645 ppm; in soil biostimulated only with *C. arietinum*, the final concentration was 11,690 ppm of WMO. In contrast, in agricultural soil polluted by WMO that was not biostimulated and was used as negative control, natural attenuation reduced the WMO to 33,531 ppm. In the soil contaminated by 65,418 ppm of WMO, the double biostimulation by sowing *C. arietinum* applied as a GM and the CFE had a positive effect, where the remaining WMO was only 42,540 ppm; in contrast, a minimal decrease in the WMO was registered in the soil that was not biostimulated (the negative control), where the final concentration of WMO was 64,500 ppm. This figure also shows that in the soil polluted by 89,533 ppm of WMO that was biostimulated by simultaneously applying the CFE and *C. arietinum*, the WMO concentration deceased to 39,366 ppm. The opposite effect was observed in the soil not biostimulated (used a negative control), with 89,200 ppm of WMO remaining. 

Figure 1B shows the profile of PHC of agricultural soil impacted by 34,5000 ppm WMO before biostimulation applied CFE and GM, where the peaks indicate the existence of a wide diversity of PHC: heterogeneous mixture of linear, branched and aliphatic PHC with 12 to 27 carbons; as well as aromatics: benzene, phenol and naphthalene. 

Figure 1C shows the profile of the disappearance of PHC in the soil impacted by 34,500 ppm of waste motor oil after biostimulation applied CFE and GM. The mineralization of linear and branched aliphatic PHC from 12 to 27 carbons, as well as aromatics such as benzene, phenol and naphthalene was detected after 30 days of biostimulation with CFE and 30 days of biostimulation used GM, due the absence of peaks supports that they were mineralized. Compared to WMO impacted soil without biostimulation, in this case due to natural attenuation mineralized in 1-methyl-4(1-methylpropyl), 2-nitro-tertari-butanol, 2-isopropyl-5-methyl heptane and octane and benzene, as well as PHC of less than 15 carbons (C ˂ 15 PHC simple linear aliphatic).

In Figure 2 biostimulation due to enzymatic activity of laccase, lignin peroxidase (LiP) and manganese peroxidase (MnP) enzymes contained in the CFE applied to biostimulate soil impacted by 34,500 ppm WMO. At first, it was shown that in the soil not impacted by WMO and not biostimulated by CFE, an activity of MnP of 5.64 IU^−1^, of 2.44 of laccase and as well as the lowest of LiP with 0.11 IU^−1^ was detected due to the absence of PHC from WMO but similar natural aromatic compounds in the agricultural soil. While the same enzymes were partially inhibited by the 34,500 ppm of WMO, in this case the soil was not biostimulated used as negative control with 6.15 IU^−1^ of MnP 0.03 IU^−1^ was detected while laccase and LiP with 0.11 IU^−1^. In the soil polluted by 34,500 ppm of WMO biostimulated applied CFE, there the activity of this monooxygenase increased probably in combination with induced free extracellular enzymes in the soil, like MnP at 30 days with 9.4, LiP with 7.16 and laccase with 1.35 IU^−1^. In soil at 60 days polluted by WMO biostimulated applied CFE, the enzyme activity was 4.96 IU^−1^ for MnP, and LiP with 3.22 IU^−1^ and for laccase 5.6 IU^−1^ due to GM biostimulation.

Figure 3 shows the phenology of *S. vulgare* inoculated with *R. irregularis* and *R. etli* during phytoremediation of an agricultural soil contaminated by 2127 ppm WMO, the phenology of *S. vulgare* was measured as: plant height (PLH) and root length (RL). *S. vulgare* in soil not polluted by WMO used as absolute control, had a PLH of 53.6 cm, a numerical value has no statistical difference, compared to *S. vulgare* in soil not polluted by WMO uninoculated with *R. irregularis* and/or *R. etli* used as relative control registered PLH of 52.9 cm. In opposite way to soil impacted by WMO where phytoremediation with *S. vulgare* was enhanced with *R. irregularis* and/or *R. etli*, there PLH was lower with 40.1 cm; in the same soil polluted by WMO with *S. vulgare* and *R. etli* a PLH with 35.0 cm was registered, while *S. vulgare* inoculated with *R. irregularis* had a PLH of 38.4 cm, both numerical values had no statistical difference. While, in the soil polluted by WMO *S. vulgare* inoculated with *R. irregularis* and *R. etli*, the PLH was 10.7 cm, in the soil *S. vulgare* inoculated only with *R. etli* registered 10.1 cm PLH. In the soil impacted with the remaining of the WMO, *S. vulgare* inoculated only with R. irregularis, had 13.4 cm RL. These RL values of *S. vulgare* inoculated only with *R. irregularis* had no statistical difference compared to *S. vulgare* used as relative control in soil not polluted by WMO, registered 14.5 cm RL.

Figure 4 shows the aerial fresh weight (AFW) of *S. vulgare* during phytoremediation an agricultural soil polluted by 34,500 ppm WMO biostimulated applied CFE and GM. It was observed that *S. vulgare* reached the highest AFW with 5.04 g in the soil not polluted by WMO used as a relative control. In the soil not impacted by WMO irrigated only with water used as absolute control, *S. vulgare* reached 4.36 g AFW. In the same figure, in the soil impacted by WMO *S. vulgare* inoculated with *R. irregularis* and *R. etli* had an AFW of 2.14 g due to the phytotoxicity of the remaining WMO. In the soil impacted by WMO *S. vulgare* inoculated only with *R. irregularis* registered 2.18 g AFW. These numerical values had no statistical difference, compared to *S. vulgare* inoculated only with *R. etli*, that reached 1.34 g AFW, the lowest registered in the trial. While *S. vulgare* in soil no polluted by WMO, used as a relative control, the radical fresh weight (RFW) reached 1.84 g. Compared to *S. vulgare* grown in soil not polluted by WMO used as absolute control that registered 0.86 g RFW, both numerical values of RFW had no statistical difference. While, in soil impacted by WMO where grown *S. vulgare* inoculated with *R. etli* and R. irregularis, 0.5 g RFW was registered. In soil impacted by WMO where grown *S. vulgare* inoculated with *R. etli* reached 0.42 g RFW. In the soil impacted by WMO where grown *S. vulgare* inoculated only with R. irregularis, registered 0.24 g RFW.

Figure 5 shows the areal and radical dry weight (ADW/RDW) of *S. vulgare* during phytoremediation of soil polluted by the remaining 2066 ppm of WMO by biostimulation applied CFE and GM. As relative control, *S. vulgare* was used in soil not polluted by WMO, and fed with 100% mineral solution, and uninoculated with *R. irregularis* and/or *R. etli* which generated an ADW of 1.6 g, followed by 1.08 of *S. vulgare* in soil not impacted by WMO irrigated only with water as an absolute control. In soil impacted by WMO, the ADW of *S. vulgare* inoculated with *R. irregularis* and/or *R. etli* reached an ADW of 0.7 g, while *S. vulgare* in soil impacted by WMO had an ADW of 0.69 with *R. irregularis* and 0.416 g with *R. etli*, compared to *S. vulgare* in soil not polluted by WMO fed with a mineral solution, uninoculated with *R. irregularis* and/or *R. etli*. This figure shows that the RDW of *S. vulgare* with 0.71 g of in soil not polluted by WMO irrigated only with water used as absolute control where this numerical value was statistically different compared to *S. vulgare* grown in soil not polluted by WMO, uninoculated used as relative control with 0.36 g of RDW. While in *S. vulgare* grown in soil impacted by WMO inoculated with *R. irregularis* and *R. etli*, registered 0.17 g RDW, a numerical value without statistical difference compared to the 0.14 g RDW of *S. vulgare* grown in soil impacted by WMO inoculated only with *R. etli*. In WMO-impacted soil, where grown *S. vulgare* inoculated only with R. irregularis, registered 0.09 g RDW.

Figure 6 shows the WMO concentration in the soil during biostimulation applied EFC and *C. arietinum* as GM and at the beginning of phytoremediation with *S. vulgare* with *R. irregularis* and *R. etli*. In the soil polluted by WMO not-biostimulated, used as negative control, the WMO concentration was reduced to 33,531 ppm after 60 days. While in the soil contaminated by WMO biostimulated applied CFE containing: laccase, MnP and LiP for 30 days, WMO was reduced to 6786 ppm. In this agricultural soil, after biostimulation applied CFE, and then *C. arietinum* as GM then WMO decreased from 6786 ppm after 30 days to 2066 ppm.

Figure 7 shows the percentage of colonization of *R. irregularis* in the root of *S. vulgare* during phytoremediation of the soil contaminated by the remaining 2066 ppm of WMO. Where it shows the percentage of colonization of native *R. irregularis* and other LMF in the root of *S. vulgare* in the soil contaminated by the remaining 2066 ppm of WMO from the biostimulation due to application of CFE and GM. The highest percentage of colonization in the root of *S. vulgare* was 79.2% of internal and external mycelium, and 10.1% of arbuscules in the soil not contaminated by WMO irrigated only with water used as absolute control. This shows that colonization of *S. vulgare* by native LMF in that soil was compared to the percentage colonization in *S. vulgare* fed with 100% mineral solution without *R. irregularis* used as relative control. A reduction in LMF colonization was detected, with 52% mycelium but no other structures were found.

Figure 8 shows the soil contaminated by 34,500 ppm of WMO biostimulated applied CFE and GM, then phytoremediated by *S. vulgare* in flowering stage with *R. etli* and *R. irregularis* to eliminate the remaining 2066 ppm of WMO up to 114 ppm, these numerical values had statistical difference compared to the soil where *S. vulgare* was grown with *R. irregularis* in this case the remaining WMO was 86.9 ppm. While in the soil impacted by WMO phytoremediated sowed *S. vulgare* inoculated with *R. etli*, WMO was reduced to 100 ppm.

Table 1 shows remnant hydrocarbons derived from mass coupled gas chromatography analysis of soil contaminated with 34,500 ppm before and after its biostimulation with a CFE, then with *C. arietinum* as GM and to conclude its depuration by phytoremediation with *S. vulgare* inoculated with *R. etli* and/or *R. irregularis*. In the soil with WMO corresponding to day 0, the PHC present before its bioremediation are shown, there was found a heterogeneous mixture of linear and branched aliphatics of 12 to 35 carbons; some aromatics such as benzene, phenols and naphthalene; it was shown that in the soil contaminated with WMO by natural attenuation considered as negative control at 120 days, phenol, 2,4-di (1,1-dimethylethyl) and branched aliphatic PHC of 12 and 14 carbons were eliminated. While it was observed that in the soil when CFE was applied during the first 30 days, aliphatic branched PHC of 12 and 14 carbons and linear PHC of up to 35 carbons and aromatics such as benzene (1-methyl-4-1-methylpropyl), 2-nitro-tertari-butanol, 2-isopropyl-5-methyl-1-heptanol were eliminated; then with the second phase of biostimulation with *C. arietinum* as GM, naphthalene, 1-cyclohex-1-1-enyl-1-phenyl-ethanol were also eliminated; and collaterally, phthalic acid di(7-methyl-octyl) ester was generated. Finally, in the third phase of phytoremediation of the same soil with *S. vulgare*, the elimination and formation of various PHC was observed: in the soil phytoremediated with this grass inoculated with *R. etli* and/or *R. irregularis*, nonadecane, eneicosane, eicosane and heptacosane were eliminated, while naphthalene and 1-cyclohex-1-1-enyl-1-phenyl-ethanol were oxidized from the aromatics. However, 1-decanol, 2-hexyl-1-, phenol, 2,4-di(1-1 dimethylethyl), heptadecane, 1-decanol and phthalic acid di(7-methyl-octyl) ester were also generated. 

## 3. Discussion

Figure 1A shows the concentration of WMO remaining in the soil impacted by 89,830, 65,418, and 34,500 ppm WMO after biostimulation applied CFE and GM. In the soil impacted by 34,500 ppm WMO, when the maximun decrease in WMO was registered, was verified by the chromatogram in Figure 1C, since a part of the PHC of WMO was mineralized, induced by biostimulation applied CFE containing MnP, laccase and LiP from CFE, that hydrolyze phenolic and non-phenolic aromatics into secondary metabolites amenable to mineralization by native heterotrophic microbial populations [22]. This was followed by soil biostimulation using GM that enriched the soil with organic C and N compounds to balance the C:N ratio for the aerobic heterotrophic microbial population to mineralize WMO short-chain aliphatic compounds [23]. GM increased the concentration of simple organic C from plants to facilitate coometabolism of other long-chain PHC from WMO [24,25,26]. In that sense, chromatogram in Figure 1B shows the PHC from WMO before biostimulation of soil impacted by 34,500 ppm WMO where the different peaks of aliphatic, aromatic and polycyclic PHC from WMO causing fertility loss and inhibition of soil organic matter mineralization due to their toxic properties [2,24,27] before being biostimulated by CFE and GM.

Figure 2 shows the enzyme activity in the soil contaminated by WMO biostimulated applied CFE, where the activity of MnP and LiP was reduced after 60 days, in opposite way laccase was increased due GM biostimulation. The concentration of the remaining WMO did not inhibit laccase activity, suggesting that synthesis by native LMF that hydrolyze lignin due biostimulation by GM. Inhibition of enzyme activity was evident in the CFE, because to the minimal availability of aromatic PHC from the WMO that were substrate on which even adverse soil physicochemical factors that reduced these enzymes activity [22,25,28]. Consequently, in WMO impacted soil, biostimulation applied CFE and GM decreased WMO concentration, to facilitate phytoremediation by *S. vulgare* enhanced with *R. etli* and *R. irregularis* that improving root plant uptake mineral, increased the phytodegradation capacity of WMO PHC and at the same time resistance to the remaining toxic PHC [28].

In Figure 3, it is suggested that because there were some PHC remaining from the WMO, for biostimulation applied CFE and GM, those PHC could be phytotoxic on the PLH of *S. vulgare*, that lyse the plasma membranes of the cortical cells of the roots, cause a higher energy expenditure in the plant, due aerial growth is limited [28,29]. Regarding the RL of *S. vulgare* when grown in soil not contaminated by WMO, irrigated only with water, was observed promoting root elongation in response to absence of essential minerals as well as phosphates (PO^−3^_4_) in the agricultural soil [30,31]. Since is well known that *R. irregularis* and *R. etli* can improve the mineral basic uptake inside the root of *S. vulgare* under stress mineral conditions [28,29]. Besides that, endomycorrhizal fungus like *R. irregularis* is able confer resistance to plants to avoid abiotic stress caused by WMO PHC. 

Figure 4 shows the response of *S. vulgare* inoculated with plant growth promoting microorganisms attributed to WMO PHC stress. Where the resistance of *S. vulgare* to WMO toxicity at the root level is reported due to a benefical plant interaction with *R. irregularis* that optimizes plant demand mineral uptake. Since+ *R. etli* and *R. irregularis* are able to hydrolyze and facilitate the mineralization of aromatic PHC from WMO [5,32,33].

Figure 5 supports that the phytotoxicity of WMO polycyclic aromatic PHC cause plant stress by blocks mineral and water uptake, that reduces biomass production [28,29]. A negative effect on *S. vulgare* growth due to phytotoxicity of WMO PHC has been reported [34]. Cheema et al. (2008) [35] reported that in a soil polluted by 70 ppm phenanthrene, 72 ppm pyrene had a negative effect on the biomass of *Festuca arundinacea* that causing a 53.5% decrease compared to when *F. arundinacea* grew in a soil not contaminated with these aromatics. Therefore, it is important to select some specific plants inoculated with plant growth-promoting microorganisms to improve phytoremediation of agricultural soils contaminated by WMO or other aliphatic and/or aromatic-type mixed PHC [33].

Figure 6 indicates that biostimulation of WMO contaminated soil by *C. arietinum* enriched the soil with organic compounds released by radical exudates [2,36] for the growth of the native microbial population. Subsequently, the incorporation of *C. arietinum* into the soil as a GM increased with more organic N compounds, to improve the balance of the C:N ratio to oxidate the WMO [22,24,25] at concentration level below to maximun accepted by the NOM-138 SEMARNAT/SS; SEMARNAT-INE, 2009 [7,37].

In Figure 7 it has also been shown that colonization by LMF such as *R. irregularis* in some plants makes them tolerant to the negative effect of aromatic PHC from WMO on their growth and accelerates soil bioremediation [5,38]. In this figure it was also observed that *S. vulgare* alone by this LMF with 59% colonization generated intra- and extraradical mycelium, arbuscules and vesicles. These LMF have been shown to be tolerant to some aromatic PHC contained in the WMO such benzene, phenol, naphthalene and others [32,39]. However, those aromatic PHC have been reported to partially inhibit the colonization of *R. irregularis* and other LMF in various plants, because they cause lysis of cell membranes, blockage of photosynthesis, lipids and proteins, consequently less radical plant tissue is generated [40]. As demonstrated in *S. vulgare* with *R. irregularis* and *R. etli*, 31% colonization by a native LMF was recorded, a percentage value that is not statistically different compared to 24% in *S. vulgare* with only *R. etli*, in both cases a higher percentage of mycelium and a lower percentage of arbuscules, because WMO PHC inhibit the synthesis of structural and reserve lipids of LMFs in plant cells, and negatively the architecture of cell membranes. While in *S. vulgare* used as absolute control the low availability of nutrients such as N, P and K stimulated native colonization and generated the different structures of this LMF. Therefore, it is important to select plants and plant growth-promoting microorganisms that are tolerant to aromatic PHC and at the same time capable of mineralizing them to complete a successful recovery of a soil impacted by mixtures of PHC such as those existing in the WMO [26,27,31].

Figure 8 shows that individual inoculation of *S. vulgare* with *R. irregularis* was the best option to conclude the recovery of that soil. In this regard, it has been reported that *R. irregularis* improves plant growth by conferring the plant resistance to abiotic stress such as that generated by the WMO, its establishment in contaminated soil, provides nutrients, improves gas exchange, absorption and assimilation of PHC by plants [41]. Also, that *R. irregularis* can absorb and assimilate some of the PHC in its hyphae, which improve the recovery of that soil [31]. In the three cases of phytoremediation soil is biorecovery, since the WMO concentration was the lowest that is detected in any agricultural soil never polluted by PHC as WMO. According to NOM-138 SEMARNAT/SS; SEMARNT-INE, 2009 mexican regulation related to [7] this soil is already bioremediated. 

The dynamic of remediation agricultural soil impacted by WMO shown in this research like all other biological techniques of remediation, involves high costs, so it cannot be seen as a business, but as a necessary and effective action for the recovery of natural resources.

In relation to the Table 1, it has been reported that EFC monooxygenases oxidize alkanes and other petroleum PHC and generate alcohols, aldehydes and carboxylic acids, mainly; MnP oxidizes phenolic aromatic compounds while laccase oxidizes non-phenolic aromatics [42]. According to the literature, the diversity profile of WMO PHC, both aliphatic and aromatic, resulting from phytoremediation is one of the first reported that shows that a part of the PHC is mineralized, another part is degraded and generates new PHC that can even be recalcitrant. Some related research has reported that phytoremediation of soils contaminated with petroleum PHC such as benzene, naphthalene, anthracene or pyrene is an option that favors the elimination of a heterogeneous mixture of PHC because the root system of these plants partly improves the soil structure favoring gas exchange and through its exudates promotes heterotrophic microbial populations to oxidize PHC [43]. It has also been reported that inoculation of plants such as *S. vulgare* with *R. irregularis* accelerates soil purification by favoring the assimilation of some PHC in plant lipid cells and also in the hyphae of this LMF [39].

## 4. Materials and Methods

Degraded sodic lateritic agricultural soil was used, from a 20-year-old field under an intensive system of crops: *Zea mays-Triticum aestivum* and *Z. mays-Hordeum vulgare*, called “La cajita” of Tenencia Zapata in the municipality of Morelia, Michoacán, Mexico, km 5 of the Morelia-Pátzcuaro highway, Mexico. A 1.0 kg of soil was used in each experimental unit called Leonard’s Jar, which was previously sieved with #40 (0.0165 in). The soil was also subjected to an analysis of the physicochemical properties according to the Mexican standard NOM-021-SEMARNAT-2000 [44], where physicochemical properties of agricultural soil were pH 6.1 moderately acidic, organic matter 7.33% very high, cation exchange capacity Cmol (+) Kg: 15 low, texture: %clay/silt/sand. 7-12-81 sandy loam, bulk density g/cm: 0.96 common in volcanic soil, porosity %: 53 slightly high, percentage (%) of moisture saturation %: 63 high, field capacity %: 8.4 low, usable moisture (%): 4.2% low. Considering that the concentration levels of PHC when a soil is contaminated are variable, this soil was contaminated with 34,500, 65,418 and 89,830 ppm of WMO that was mixed with a 1.0% commercial detergent to emulsify the insoluble mixture and facilitate its availability to mineralization [45].

Biostimulation of WMO contaminated soil by a crude fungal extract followed by *Cicer arietinum* as green manure.

The experiment was conducted in Leonard’s Jars, as shown in Figure 9, was maintained in a greenhouse, using a randomized block design with five replications. In that impacted by 34,500, 65,418, 89,830 ppm of WMO. The WMO was from a mechanical workshop in the city of Morelia, Michoacán, Mexico, whose chromatographic analysis is shown in Figure B of the results section. In each Leonard’s Jar with impacted soil, it was first biostimulated for 30 days by CFE, and then the activity of the extracellular enzymes contained in the CFE, such as laccase, lignin peroxidase and manganese peroxidase, was measured. For enzyme measurement, 30 g of this soil was collected and kept until analysis at 4 °C, by taking 1.0 g which was diluted in 9.0 mL of 0.85% NaCl-5% Tween 80 (100:1). The mixture was kept under agitation for 10 min/150 rpm and centrifuged at 8000 rpm/4 °C/15 min [46]. Biostimulation was followed by sowing *C. arietinum* as GM where the seed was germinated in a moist, sterile cotton bed under dark conditions, after emergence it was transplanted into Leonard’s Jars containing the soil previously biostimulated by CFE; 30 days after sowing *C. arietinum* was cut into pieces of approximately 0.5 cm and incorporated for 30 days into the soil, which was moistened to field capacity [47]. The experiment also consisted of an absolute control soil: soil not polluted by WMO irrigated with water and a negative control soil: WMO-contaminated soil without biostimulation.

Soil phytoremediation to mineralize remaining WMO with *Sorghum vulgare* co-inoculated with *Rhizophagus irregularis* and/or *Rhizobium etli*.

For the soil phytoremediation stage, only the result of biostimulation of soil impacted by 34,500 ppm WMO with CFCs and GM was considered, because the rate of WMO reduction was lower than the maximum value of NOM-138 SEMARNAT/SS [7].

Soil phytoremediation at Leonard’s Jars was performed according to the experimental design (Figure 10) to biorecover an agricultural soil impacted by remain of WMO. 

It consisted of absolute control: soil not polluted by WMO with *S. vulgare* uninoculated with *R. irregularis* or *R. etli*, irrigated with water only; a relative control: soil not polluted by WMO with *S. vulgare* uninoculated with *R. irregularis* or *R. etli* fed with 100% mineral solution (MS); a negative control: soil contaminated by WMO non biostimulated (CFE and/or GM) and non-phytoremediated with *S. vulgare* neither inoculated with *R. irregularis* + *R. etli*, such controls were used to comparison with soil pollutes by WMO non biostimulated and phytoremediated; treatment 1 soil polluted by WMO biostimulated, sowed with *S. vulgare* with *R. irregularis* and *R. etli* fed with 50% MS; treatment 2 soil polluted by WMO biostimulated, sowed with *S. vulgare* with *R. etli* + 50% MS; treatment 3 soil polluted by WMO sowed with *S. vulgare* inoculated with *R. irregularis* + 50% MS. *R. etli* and *R. irregularis* were taken from the microbial collection of the Environmental Microbiology Laboratory of the IIQB of the UMSNH. These strains were preserved in vials with sterile soil. The response variables were phenology and biomass at the physiological level of flowering; as well as the percentage of colonization of *R. irregularis* in its roots after 120 days. Finally, the initial and final concentration; and the identification of WMO PHC [48]. The experimental data were subjected to a combined analysis of variance (ANOVA) and Tukey *p* < 0.05 (JMP program version 6.0).

WMO extraction from agricultural soil biostimulated and phytoremediated.

Soil impacted by WMO samples were stored at 4 °C, dried at room temperature for 48 h, homogenized and three 3.0 g aliquots of each were taken for a total of 3 replicates. They were placed in 10.0 mL hash tubes and 3.0 mL of hexane (purity 98.5% ACS-Sigma Aldrich HPLC grade) was added; it was stirred for 60 s in bortex and subsequently filtered through a separating funnel three times; water residues were removed in a funnel with a cotton filter and MgSO_4_, finally the solvent was allowed to evaporate for 24 h at room temperature and each sample was calibrated to a total volume of 1.0 mL for subsequent analysis [49].

Identification of hydrocarbons of WMO by gas chromatography coupled to mass.

The soil samples biostimulated polluted by WMO were analyzed in an Agilent Technologies 7890A series gas chromatograph coupled to 5975C series masses, 1.0 µL of sample was injected in Spitless mode, helium (99.995% purity) was used as carrier gas in a Zebron capillary column. 5 MS 30.0 m long, with an internal diameter of 0.25 mm and a film thickness of 0.25 mm. The injector temperature was 250 °C and the initial oven temperature was 50 °C with a ramp of 30–150 °C, ramp 2 of 10 °C min up to 310 °C/2 min. The temperature detector was 280 °C with an equilibration time of 3 min and a maximum temperature of 320 °C (modified method of Peng et al., 2009) [49]. 

The experimental data were subjected to a combined analysis of variance (ANOVA) and Tukey < 0.05 (JMP program version 6.0).

Figure 11 shows the biorecovery of an agricultural soil impacted by WMO biostimulated applied CFE and *C. arietinum* as GM, due to decrease concentration of WMO, facilitates phytoremediation sowing *S. vulgare* inoculated with *R. irregularis and/ R. etli* to reduce the WMO at concentration value equivalent to that detected in a soil not contaminated by mixtures of PHC as WMO type [26,45].

## 5. Conclusions 

This work demonstrated the potential of different biological actions for the remediation of a contaminated soil. In which previously mechanical actions must be performed to eliminate the excess of hydrocarbons. Subsequently, the biological action through biostimulation drastically reduces the concentration of toxic hydrocarbons in the WMO of the soil. This drastic reduction of the WMO, allows the action of phytoremediation by means of plants to eliminate the remaining WMO to a similar level of a soil that was never contaminated. From the above, it is concluded that the soil WMO bioelimination strategy was effective, environmentally friendly and safe for the reestablishment of the plant crop for human and plant health.

## Figures and Tables

**Figure 1 plants-12-01053-f001:**
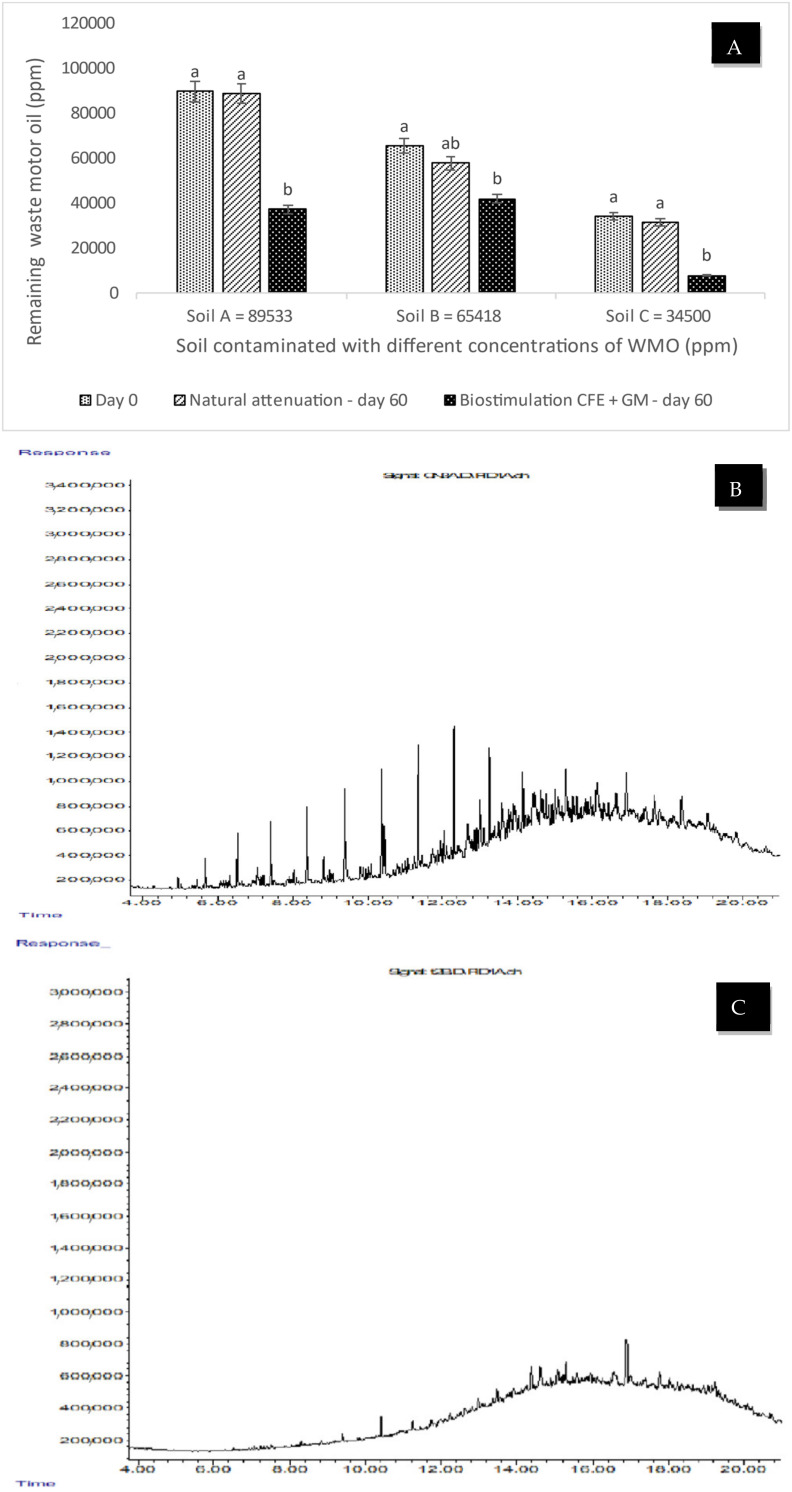
Concentration of remaining waste motor oil in agricultural soil biostimulated applied crude fungal extract and *Cicer arietinum* as green manure (**A**). Chromatogram of hydrocarbons from waste motor oil before biostimulation of soil (**B**). Chromatogram of the disappearance of hydrocarbons in the soil impacted by 34,500 ppm of waste motor oil after biostimulation using crude fungal extract and *Cicer arietinum* as a green manure (**C**). Notes: CFE = crude fungal extract, GM = green manure. Different letters indicate statistical difference according to ANOVA-Tukey (*p* < 0.05). Number of repetitions (*n*) = 5.

**Figure 2 plants-12-01053-f002:**
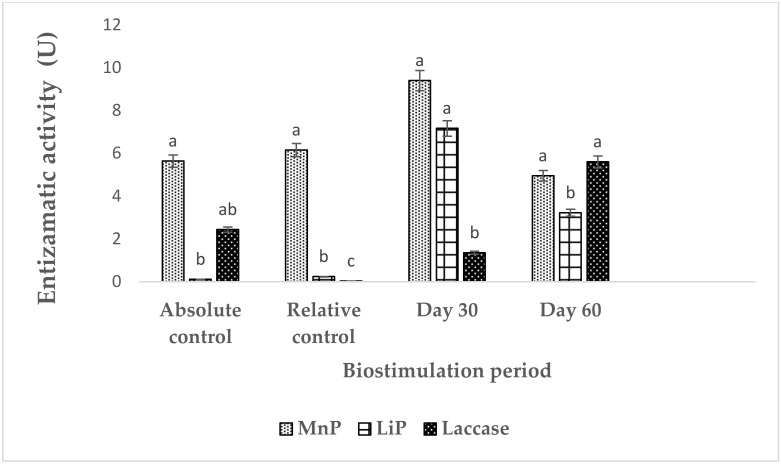
Enzymatic activity in soil impacted by 34,500 ppm of waste motor oil biostimulated applied crude fungal extract and Cicer arietinum as a green manure. Notes: LiP = Lignin peroxidase; MnP = Manganese peroxidase, CFE = crude fungal extract, GM = green manure, MISO = mineral solution, WMO = waste motor oil. Different letters indicate statistical difference according to ANOVA-Tukey (*p* < 0.05). Number repetitions (*n*) = 5.

**Figure 3 plants-12-01053-f003:**
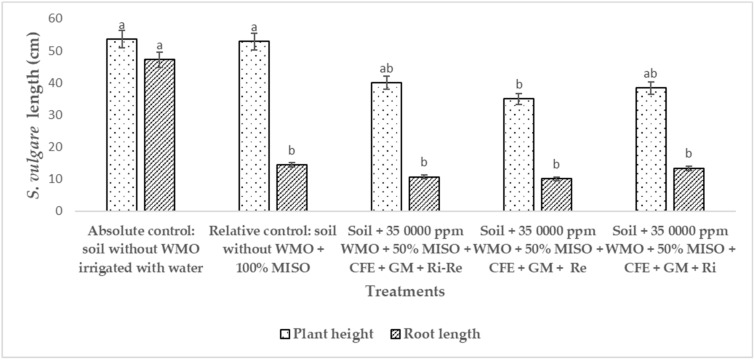
Phenology of *Sorghum vulgare* inoculated with *Rhizobium etli* and *Rhizophagus irregularis* after 120 days of phytoremediation of soil polluted by 2127 ppm WMO remaining of waste motor oil. Notes: LiP = Lignin peroxidase; MnP = Manganese peroxidase, CFE = crude fungal extract, GM = green manure, MISO = mineral solution, Ri = *Rhizophagus irregularis*, Re = *Rhizobium etli*, WMO = waste motor oil. Different letters indicate statistical difference according to ANOVA-Tukey (*p* < 0.05). Number repetitions (*n*) = 5.

**Figure 4 plants-12-01053-f004:**
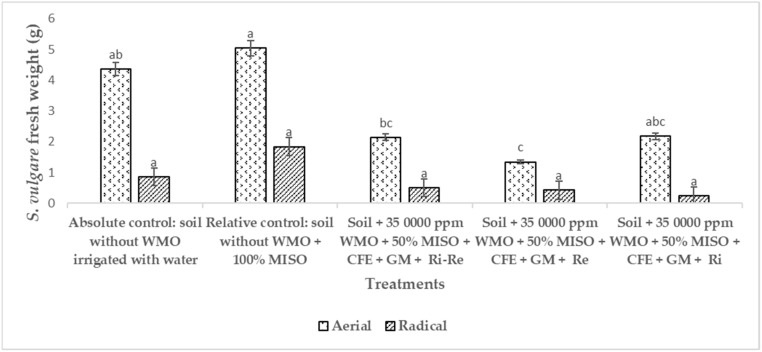
Aerial and radical fresh weight of *Sorghum vulgare* inoculated with *Rhizobium etli* and *Rhizophagus irregularis* after 120 days of phytoremediation of soil polluted by remaining waste motor oil. Notes: CFE = crude fungal extract, GM = green manure, MISO = mineral solution, Ri = *Rhizophagus irregularis*, Re = *Rhizobium etli*, WMO = waste motor oil. Distinct letters indicate statistical difference according to ANOVA-Tukey (*p* < 0.05). Number of repetitions (*n*) = 5.

**Figure 5 plants-12-01053-f005:**
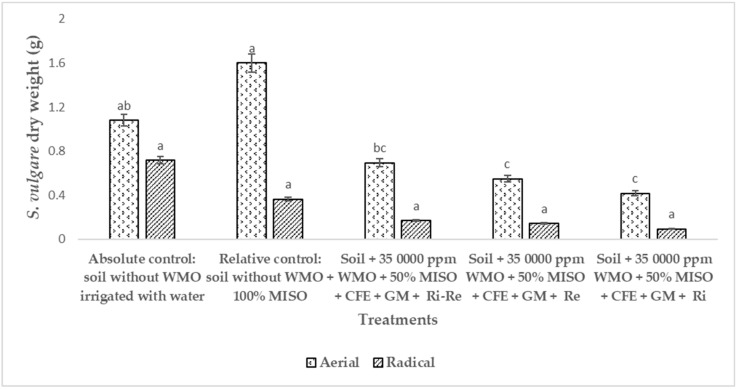
Dry aerial and radical weight of *Sorghum vulgare* with *Rhizobium etli* and/or *Rhizophagus irregularis* after 120 days of phytoremediation of polluted by waste motor oil. Notes: CFE = crude fungal extract, GM = green manure, MISO = mineral solution, Ri = *Rhizophagus irregularis*, Re = *Rhizobium etli*, WMO = waste motor oil. Different letters indicate statistical difference ANOVA-Tukey (*p* < 0.05). Number of repetitions (*n*) = 5.

**Figure 6 plants-12-01053-f006:**
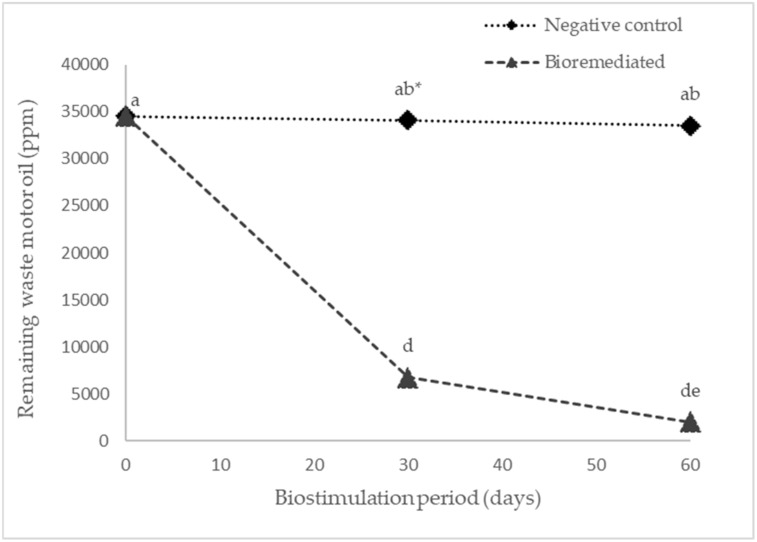
Concentration of soil polluted by 34,500 ppm of waste motor oil biostimulated applied crude fungal extract and Cicer arietinum as a green manure and start to phytoremediation by *Sorghum vulgare* with *Rhizophagus irregularis* and/or *Rhizobium etli*. * Different letters indicate statistical difference ANOVA-Tukey (*p* < 0.05). Number of repetitions (*n*) = 5.

**Figure 7 plants-12-01053-f007:**
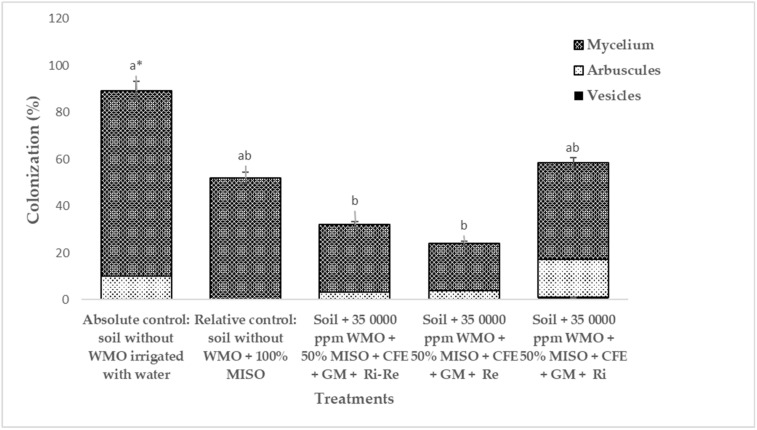
Colonization percentage of *Rhizophagus irregularis* on roots of *Sorghum vulgare* at flowering stage during phytoremediation of soil polluted by remained of 2066 ppm of waste motor oil. Notes: CFE = crude fungal extract, GM = green manure, MISO = mineral solution, Ri = *Rhizophagus irregularis*, Re = *Rhizobium etli*, WMO = waste motor oil. * Distinct letters indicate statistical difference ANOVA-Tukey (*p* < 0.05). Number of repetitions (*n*) = 5.

**Figure 8 plants-12-01053-f008:**
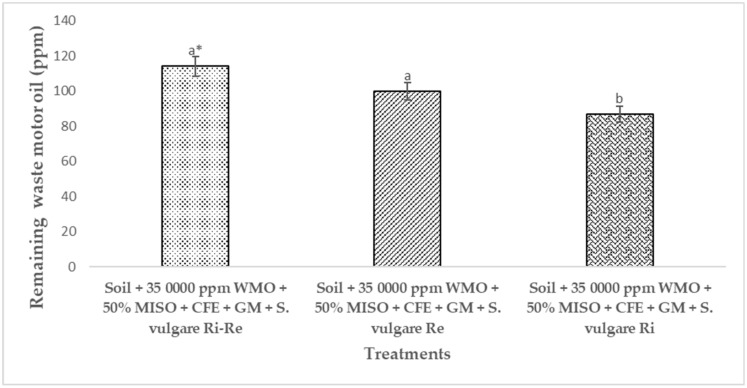
Concentration of waste motor oil in soil after 120 days of phytoremediation by *Sorghum vulgare* inoculated with *Rhizophagus irregularis* and/or *Rhizobium etli*. Notes: CFE = crude fungal extract, GM = green manure, MISO = mineral solution, Ri = *Rhizophagus irregularis*, Re = *Rhizobium etli,* WMO = waste motor oil. * Distinct letters indicate statistical difference ANOVA-Tukey (*p* < 0.05). Number of repetitions (*n*) = 5.

**Figure 9 plants-12-01053-f009:**
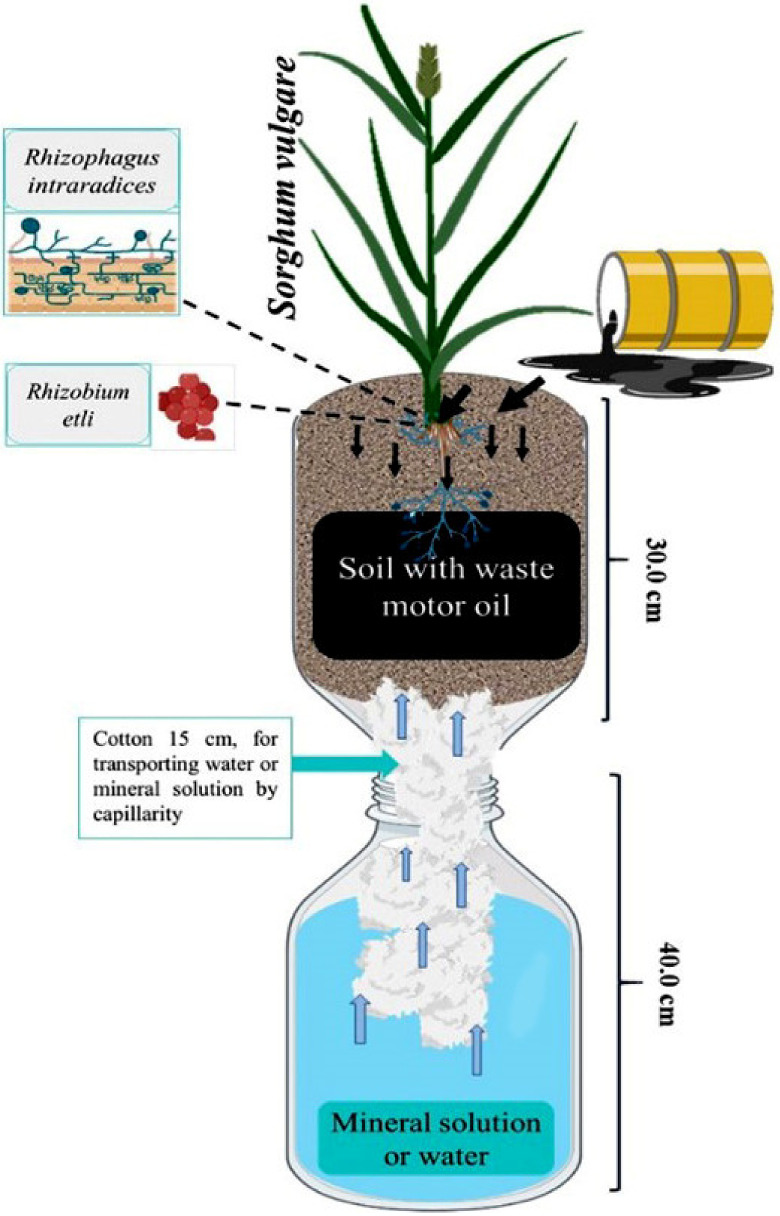
Leonard’s Jar for phytoremediation of agricultural soil polluted by WMO.

**Figure 10 plants-12-01053-f010:**
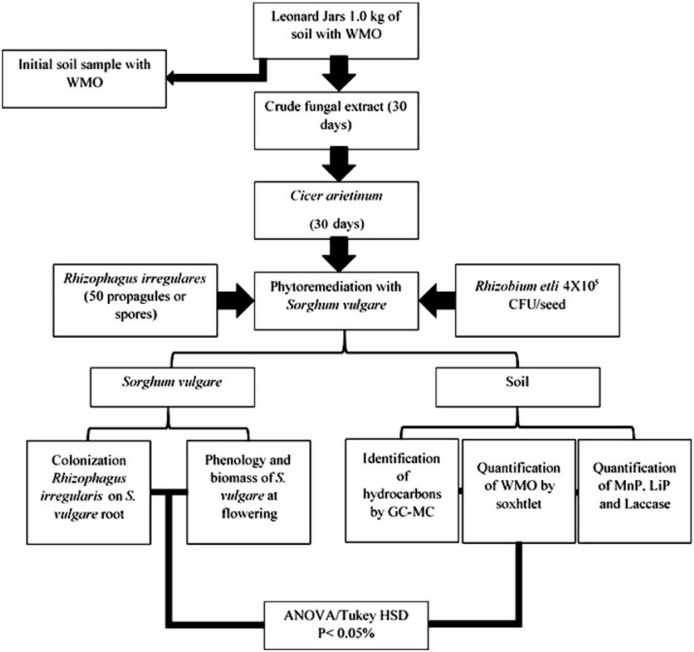
Experimental design of biostimulation of agricultural soil impacted by WMO applied crude fungal extract then with *C. arietinum* and phytoremediation of remaining of 34,500 ppm of WMO by *Sorghum vulgare* inoculated with *Rhizophagus irregularis* and/or *Rhizobium etli.* Notes: WMO = waste motor oil; GC-MC = gas chromatography coupled to mass; LiP = lignin peroxidase; MnP = manganese peroxidase; CFU = colony forming units.

**Figure 11 plants-12-01053-f011:**
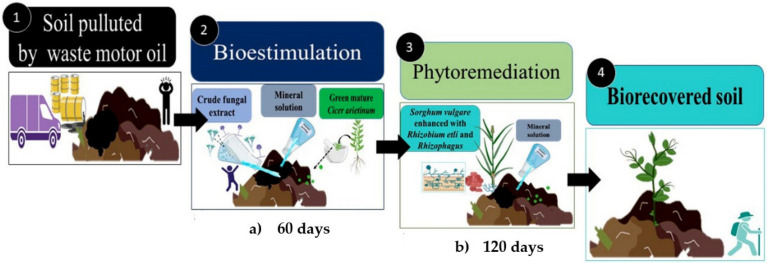
Scheme of the Biorecovering of an agricultural soil impacted by waste motor oil, biostimulated applied a crude fungal extract and *Cicer arietinum* as green manure followed for phytoremediation using *Sorghum vulgare* inoculated with *Rhizobium etli* and *Rhizophagus irregularis*.

**Table 1 plants-12-01053-t001:** Remnant hydrocarbons derived from mass coupled gas chromatography analysis of soil with 34,500 ppm of waste motor oil before and after biostimulation with a crude fungal extract, *Cicer arietinum* and phytoremediation with *Sorghum vulgare* inoculated with *Rhizophagus irregularis* and/or *Rhizobium etli*.

Identified Hydrocarbons	Chemical Formula	Chemical Structure	Start: Soil + WMO Day 0	Negative Control-Day 120	BiostimulationDays 30/30	Phytoremediation with *S. vulgare* Day 120
Crude Fungal Extract	*C. arietinum* as Green Manure	*R. irregularis* y *R. etli*	*R. etli*	*R. irregularis*
Aromatics	C_12_H_12_	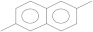	+	+	+	-	-	-	-
C_14_H_22_O	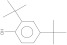	-	-	-	-	+	+	+
C_14_H_18_O	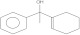	+	+	+	-	-	-	-
C_24_H_38_O_4_	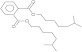	+	+	+	+	-	-	-
C_26_H_42_O_4_	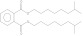	-	-	+	+	+	+	+
Linear aliphatics	C_17_H_36_	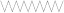	+	+	+	+	-	+	+
C_21_H_44_	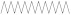	+	+	+	+	-	-	-
C_27_H_56_	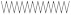	+	+	+	-	-	-	-
C_35_H_70_	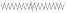	+	+	+	-	-	-	-
Branched aliphatics	C_12_H_24_	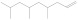	+	+	+	+	-	-	-
C_13_H_28_	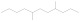	+	+	+	+	-	-	-
C_17_H_36_O	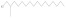	+	+	+	+	+	+	+
C_21_H_44_	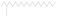	+	+	+	+	+	-	-

Notes: WMO = waste motor oil; sign (+) = presence of the hydrocarbon; (-) absence of the hydrocarbon. Result generated by comparison in NIST MS 2 Library.

## Data Availability

Not applicable.

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
