# Peer review of "Bioelimination of Phytotoxic Hydrocarbons by Biostimulation and Phytoremediation of Soil Polluted by Waste Motor Oil"

_plants, 2023, doi:10.3390/plants12051053_

Round 1
Reviewer 1 Report (Previous Reviewer 4)
The manuscript has been greatly improved. Just a few spelling errors:
Table 1) Linear aliphatics .... C21H4 ?
Line 333) no new paragraph
Line 438) Stile .... Biostimulation was followed by sowing C arietinum ...
Line 439) .... dark. After ....
Reference 9 has been exchanged - intentionally?
Author Response
Please see the attachment

Reviewer 2 Report (Previous Reviewer 2)
Reviewer
MDPI – Plants
Manuscript Number: plants- 2142597
Title: « Bioelimination of phytotoxic hydrocarbons by biostimulation and phytoremediation of soil polluted by waste motor oil ».
The word "Bioelimination" occurs only in the title in fig. 11. It is necessary to clarify the meaning of this term in the introduction.
line 10 The abstract does not contain an introductory word about the importance of the study. The authors immediately describe the tasks that they solved in the course of the study.
line 23 In the annotation, you need to indicate the further direction of using the results of the study.
line 29 This phrase should be abbreviated in full: petroleum hydrocarbon (PHC)
line 78 Research goal missing.
line 450 9 it is necessary that the names of all parts of the figure be turned in one direction so that the reader does not turn his head and does not get tired.
line 483 "Symbology:" should be replaced with "Notes:"
line 524 In the "Conclusion" section, the main conclusions of the article should be included, so it needs to be supplemented.
In the list of references, many sources omitted authors, and there are other design errors.

Author Response
Please see the attachment

Reviewer 3 Report (Previous Reviewer 3)
Speaking about work in general, there are several key points, the solution of which seems to be an insurmountable obstacle.
1) The quality of the presented material. For example, Figure 1A. The impression is that they took a low-resolution drawing and stretched it in a graphic editor. They did the same with the others.
2) We cannot get unambiguous answers to the questions from the authors:
a) about the characteristics of the pollutant (full composition and description). Instead, the authors refer to the "chromatogram" of the contaminated soil extract.
b) about the reasons for choosing concentrations. The authors cannot specifically answer the question posed. A brief example from the answers:
Point 6. Detailed answers to questions have not been given (ignored) and comments that were made earlier have not been corrected (indicated in italics): What is the reason for choosing exactly these concentrations of oil for contamination and the duration of recultivation procedures with different components?
Response 6. The agricultural soil was contaminated in a discretionary way with the WMO because generally caused or accidental hydrocarbon spills happens like this. While the duration of the bioremediation experiment was based on previous results of biostimulation of soil impacted by WMO through the application of essential minerals to induce the aerobic heterotrophic aerobic microbiota to oxidize them, while the application of fungal extracellular enzyme extract that are stable to the conditions environmental conditions of the soil, which allows the hydrolysis and subsequent elimination of aromatic hydrocarbons.
Point 7. When working with soils with inducted pollution, concentrations already available in practice are taken as a reference. For example, at landfills, with registered cases of pollution, etc.
Response 7. When agricultural soil is contaminated with mixtures of hydrocarbons such as WMO, it is impossible to initiate bioremediation with a biological action, since the high concentration of extremely toxic hydrocarbons prevents it. Therefore, the cleaning of the soil begins with a mechanical removal until a level of reduction of the WMO concentration that first allows the biostimulation that decreases the concentration to a level for sowing of plants that tolerate it and that can be more effective for the elimination. definition of the WMO as microorganisms that promote plant growth, the result is a concentration of hydrocarbons similar to or less than what is naturally detected in soil not contaminated with hydrocarbons for the reuse of soil for agricultural production that is safe for human consumption. and/or animal. For which the successful results of experiments carried out in soil with different concentrations of WMO, types of biostimulation and phytoremediation are used.
3) which strains are used in this work, under which collection numbers are they deposited? The article only indicates where the strains were obtained from, but the collection numbers of the strains were never provided to us.
Summarizing the above, the publication of the article in this form is impossible. And in the newly submitted work for review, the previously repeatedly expressed comments were not eliminated, despite the fact that it is not difficult to correct the presentation of the results in the presence of raw data.
Round 2
Reviewer 3 Report (Previous Reviewer 3)
The images do not meet the editorial requirements. Namely: File for Figures and Schemes must be provided during submission in a single zip archive and at a sufficiently high resolution (minimum 1000 pixels width/height, or a resolution of 300 dpi or higher). Common formats are accepted, however, TIFF, JPEG, EPS and PDF are preferred. The peaks on the chromatograms are not identified either by standards or by retention time. I.e. it is impossible to say unequivocally which components of the mixture correspond to the peaks on the chromatograms. The table presented in the manuscript also does not clarify this point. The article does not present the strain number in the collection of microorganisms. Describe the strain used in the work: In which organization is it supported? By whom, when and from where the strain was obtained. What number was assigned to this strain?Author Response
The remarks and comments have been answered in the attached file, thanks for the wait

This manuscript is a resubmission of an earlier submission. The following is a list of the peer review reports and author responses from that submission.
Round 1
Reviewer 1 Report
This research studied the effect of Biostimulation, and phytoremediation of agricultural soil impacted by waste residual oil: main hydrocarbons involved. the paper has numerous flaws that make the manuscript unsuitable to be published in the plant. Some of my comments are as below:
· The abstract readability should be improved
Is there any control to compare it with biosimulation and phytoremediation?
· Introduction: Overall, the introduction is useless and needs more justification to show the novelty I recommend the authors introduce and add a few paragraphs about the global challenge of pollution, the health effect of toxic materials, their treatment techniques, and the role of d green manure as stimulant and understanding of soil biology and the natural partnerships existing between the various living organisms occurring in soil (both plants and microbiota) is leading to a growing interest in the application possibilities of adaptive biological dynamics in remediation
· Why is bioremediation better than other physical-chemical processes?
· Why are single remediation methods often limited and may be disturbed by environmental conditions in petroleum hydrocarbon-contaminated soils?
· Why is coupling biostimulation and phytoremediation for the restoration of petroleum hydrocarbon-contaminated soil more necessary than others?
· Why is lignin used more than another aromatic-based biopolymer?
· How did you select the optimal strains?
· To make the paper stronger and novel the authors must include in their data about
(1) optimizing the single-factor (moisture content, the leavening agent content, and the compound fertilizer content) biological stimulation experiment,
(2) selecting plants with high tolerance to petroleum hydrocarbons,
(3) designing the artificial biostimulation-phytoremediation combined remediation experiment.
Therefore, the result of this study would provide valuable information for increasing the efficiency of the TPH degradation rate
· The data analysis should be improved.
· Ensure you predefined all abbreviated words before use.
· The authors need to pay attention to extensive futurology in a separate section with more and deep descriptions.
· Could you please consider an economic cost analysis of the process developed, to observe whether its use on a full scale is viable?
· Please discuss the environmental feasibility of combined biosimulation and phytoremediation
· In addition to this, the authors have some similarities; they published three months ago in the same journal.
· https://www.mdpi.com/2223-7747/11/11/1419/htm
In short, the manuscript was written in poor English and not well-structured so it is not worth going through to the end. Its novelty was not enough and too many errors existed. Therefore, I recommend the rejection of the manuscript.
Author Response
Response to Reviewer 1 Comments
Point 1: The abstract readability should be improved
Response 1: The readability of the summary was improved in lines 8-27.
Point 2: Is there any control to compare it with biosimulation and phytoremediation?
Response 2: The description of the controls are described in the materials and methods section: lines 400-402, and 415-418.
Point 3: Introduction: Overall, the introduction is useless and needs more justification to show the novelty I recommend the authors introduce and add a few paragraphs about the global challenge of pollution, the health effect of toxic materials, their treatment techniques, and the role of d green manure as stimulant and understanding of soil biology and the natural partnerships existing between the various living organisms occurring in soil (both plants and microbiota) is leading to a growing interest in the application possibilities of adaptive biological dynamics in remediation.
Response 3: The entire introduction section has been modified with the requested indications
Point 4: Why is bioremediation better than other physical-chemical processes?
Response 4: It is described on lines 50-55 in the introduction.
Point 5: Why are single remediation methods often limited and may be disturbed by environmental conditions in petroleum hydrocarbon-contaminated soils?
Response 5: Described in the introduction in lines 56-59
Point 6: Why is coupling biostimulation and phytoremediation for the restoration of petroleum hydrocarbon-contaminated soil more necessary than others?
Response 6: Described in the introduction on lines 58-81
Point 7: Why is lignin used more than another aromatic-based biopolymer?
Response 7: Because lignin has a complex and recalcitrant composition, which may be similar to the more complex polycyclic aromatic hydrocarbons of the WMO.
Point 8: How did you select the optimal strains?
Response 8: Described on lines 76-81
Point 9: To make the paper stronger and novel the authors must include in their data about: optimizing the single-factor (moisture content, the leavening agent content, and the compound fertilizer content) biological stimulation experiment.
Response 9: Has been described in the materials and methods section.
Point 10: Selecting plants with high tolerance to petroleum hydrocarbons
Response 10: Described on lines 70-73
Point 11: Designing the artificial biostimulation-phytoremediation combined remediation experiment.
Response 11: Has been described in the materials and methods section.
Point 12: Therefore, the result of this study would provide valuable information for increasing the efficiency of the TPH degradation rate: The data analysis should be improved.
Response 12: The analysis of the results has been improved in the "Results" section.
Point 13: Ensure you predefined all abbreviated words before use.
Response 13: Verified that all abbreviations have been predefined
Point 14: The authors need to pay attention to extensive futurology in a separate section with more and deep descriptions.
Response 14: It is not possible for us to make statements about the future, we can only defend the experiment based on the evidence obtained, otherwise we would be speculating.
Point 15: Could you please consider an economic cost analysis of the process developed, to observe whether its use on a full scale is viable?
Response 15: The dynamic of remediation agricultural soil impacted by WMO shown in this research like all other biological techniques of remediation, involves high costs, so it cannot be seen as a business, but as a necessary and effective action for the recovery of natural resources.
Point 16: Please discuss the environmental feasibility of combined biosimulation and phytoremediation
Response 16: Described on lines 50-81
Point 17: In short, the manuscript was written in poor English and not well-structured so it is not worth going through to the end. Its novelty was not enough and too many errors existed
Response 17: All the wording and structure were already modified and improved.

Reviewer 2 Report
Reviewer
MDPI - plants
Manuscript Number: plants-1912027
Title: « Biostimulation and phytoremediation of an agricultural soil impacted by waste residual oil: main hydrocarbons involved».
The topic voiced in the article is very relevant and in demand in the modern world. Levels of soil pollution by oil and oil products are only increasing every year.
The hatching between the options in Fig. 1, 2, 3, 4, 5, 7 should be made more contrasting.
The conclusion should not contain references to any sources, only a concise conclusion based on the results of the study.
Author Response
Point 1: The hatching between the options in Fig. 1, 2, 3, 4, 5, 7 should be made more contrasting.
Response 1: The shading of the figures has already been modified.
Point 2: The conclusion should not contain references to any sources, only a concise conclusion based on the results of the study.
Response 2: The conclusion has been modified and does not contain references.

Reviewer 3 Report
The problem in the article submitted for review is global and relevant. But its solution is not presented, the design of the experiment and the presentation of the results leaves a lot to be desired.
What kind of waste residual oil was used, its composition is not presented.
What is the reason for choosing exactly these concentrations of oil for contamination and the duration of recultivation procedures with different components?
The paper does not present data on the mineralization of hydrocarbons by structure (only words, speculation).
Figure 1 is not informative, low-quality. Figure 1c – what kind of component is left? It seems that it is not less than the sum of the peaks of the other components, which are shown in Fig. 1b.
Lines 261-287 describe Fig. 1, but they are not correlated with each other.
Lines 318-327 state the toxicity of aromatic hydrocarbons of engine oil, the article deals with anthracene, phenanthrene, pyrene (this is indisputable), but are they in the oil under study? (this data is not available).
Lines 358-369 are not confirmed by visual results (chromatography, for example), a set of allegedly deleted components is presented.
Less than 50% of the links are fresh (less than 10 years old).
Why does Figure 1 show data for 60 days, and Figure 2 shows data for 90 days?
Page 4. Correct "pH" to "PH"
For what reason was this particular method of pollutant destruction chosen?
Give the characteristics of the strains of microorganisms used in the work, where was it taken from and where was it deposited?
Author Response
Point 1: The problem in the article submitted for review is global and relevant. But its solution is not presented, the design of the experiment and the presentation of the results leaves a lot to be desired.
Response 1: All the wording and structure were already modified and improved.
Point 2: What kind of waste residual oil was used, its composition is not presented.
Response 2: The origin of the WMO is specified in the materials and methods section, lines 389-391. As well as in the results section the initial composition is shown.
Point 3: What is the reason for choosing exactly these concentrations of oil for contamination and the duration of recultivation procedures with different components?
Response 3: These concentrations were chosen in a discretionary manner, taking into account that when soil contamination by hydrocarbons occurs, it is done deliberately without accounting for the exact concentration. While the duration of the experiment was made taking into account the results of previous experiments where the effect of essential minerals and enzymes from different sources has been shown.
Point 4: The paper does not present data on the mineralization of hydrocarbons by structure (only words, speculation).
Response 4: The results on the mineralization of hydrocarbons have been shown in the chromatograms in the results section where the initial and final hydrocarbons can be seen.
Point 5: Figure 1 is not informative, low-quality. Figure 1c – what kind of component is left? It seems that it is not less than the sum of the peaks of the other components, which are shown in Fig. 1b.
Response 5: Unfortunately, it was not possible to improve the quality of Figure 1. However, in the results and discussion section, a description of the initial hydrocarbons and those that disappeared after the experiment is given.
Point 6: Lines 261-287 describe Fig. 1, but they are not correlated with each other.
Response 6: This section has already been improved and modified
Point 7: Lines 318-327 state the toxicity of aromatic hydrocarbons of engine oil, the article deals with anthracene, phenanthrene, pyrene (this is indisputable), but are they in the oil under study? (this data is not available).
Response 7: That section has already been modified in the lines 338-340, describing the hydrocarbons contained in the WMO of the experiment.
Point 8: Lines 358-369 are not confirmed by visual results (chromatography, for example), a set of allegedly deleted components is presented
Response 8: The results described in the results section on the disappearance of hydrocarbons are based on the results obtained from the chromatogram shown in the same section.
Point 9: Less than 50% of the links are fresh (less than 10 years old).
Response 9: Most of the references have been updated.
Point 10: Why does Figure 1 show data for 60 days, and Figure 2 shows data for 90 days?
Response 10: The times have already been homogenized in the figures.
Point 11: Page 4. Correct "pH" to "PH"
Response 11: The acronym pH was changed to PLH.
Point 12: For what reason was this particular method of pollutant destruction chosen?
Response 12: Described in the introduction on lines 47-81
Point 13: Give the characteristics of the strains of microorganisms used in the work, where was it taken from and where was it deposited?
Response 13: Described in the Materials and methods on lines 425-428

Reviewer 4 Report
The problem is relevant. Please do a complete re-writing of the text, and consult a person, who has more profound knowledge in english language. Use complete sentences, and avoid non-logical propagations of the text. Material and methods has to be put before the results.
Why did you perform you experiments with 7,8 - 14,9 - 20,4 fold level of the given threshold, and not close to the threshold itself?
A reshape of your introduction text (just for language correction; please reshape):
Soil contamination by petroleum hydrocarbons (HCs) such as waste motor oil (WMO) is an economical and environmental worldwide problem. Diversity of HCs involved in WMO causes a serious environmental problem and damage to any living being. This has to be eliminated by ecological strategies [1-3]. WMO is a hazardous waste, a heterogenous mixture of linear or branched aliphatic, aromatic and polycyclic HCs [1,3,4]. There is minimal information about the HCs, which mainly exist in the WMO [5,6] with respect to the toxicity for human, animal and plant life, as well as the type of HCs in the WMO that are eliminated for soil biorestoration, aiming at the production of healthy and safe plants. A report about the symbiosis of Rhizobium tropici with Leucena leucocephala indicated that monoaromatics, high molecular weight diaromatics, and polycyclic aromatics are mainly involved, when WMO pollutes the soil [14]. WMOs in the soil prevent gas exchange, mineralization of organic plant matter and thus cause loss of fertility [10,15]. When the concentration of HCs from the WMO is greater than 4400 ppm, it is an environmental problem in Mexico according to the Mexican standard NOM-138-SEMARNAT/SS [4]. Its removal requires a biologically relatively easy way. Therefor, the objectives of this work were the biostimulation of soils contaminated by 34500, 65418 und 89830 ppm of WMO with crude fungal extract of lignine-degrading fungi to hydrolyze aromatic compounds of WMO followed by Cicer arietinum as green manure application. This was followed by phytoremediation with Sorghum vulgare plus Rhizophagus irregularis and/or Rhizobium etli to reduce the remaining concentration of 34500 ppm of WMO to one equal to NOM-138 SEMARNAT/SS [4,16] or less that that detected naturally, in order to allow its reuse in the production of safe food for humans and animals.
Lines 48-63 are double and have to be deleted. The second versions seems better.
Spelling mistakes in the text explaining figure 1!
Do not use the abbreviation pH for plant height
Author Response
Point 1: The problem is relevant. Please do a complete re-writing of the text, and consult a person, who has more profound knowledge in english language. Use complete sentences, and avoid non-logical propagations of the text. Material and methods has to be put before the results.
Response 1: The entire text has been rewritten and improved. According to the guidelines of the journal, it establishes that the material and methods section should be placed in the last section after the discussion.
Point 2: Why did you perform you experiments with 7,8 - 14,9 - 20,4 fold level of the given threshold, and not close to the threshold itself?
Response 2: These concentrations were chosen in a discretionary manner, taking into account that when soil contamination by hydrocarbons occurs, it is done deliberately without accounting for the exact concentration.
Point 3: A reshape of your introduction text (just for language correction; please reshape):
Response 3: The introduction section has been rewritten
Point 4: Lines 48-63 are double and have to be deleted. The second versions seems better.
Response 4: Excess lines have been deleted and the second version has been left.
Point 5: Spelling mistakes in the text explaining figure 1!
Response 5: These errors have been corrected
Point 6: Do not use the abbreviation pH for plant height
Response 6: The acronym pH was changed to PLH.

Round 2
Reviewer 3 Report
65: Error in Mn3 designation instead of Mn3+
78: Is PO4 the ionic form of a substance?
149: Dot after “Figure 3", correct to Figure 3 shows.
159: S. vulgare correct to S. Vulgare (in full italics).
253: In the phrase "In the figure 8 shows" correct English (this applies to the entire text. For example, there is a similar inaccuracy in the Discussion section).
Detailed answers to questions have not been given (ignored) and comments that were made earlier have not been corrected (indicated in italics):
What is the reason for choosing exactly these concentrations of oil for contamination and the duration of recultivation procedures with different components?
When working with soils with inducted pollution, concentrations already available in practice are taken as a reference. For example, at landfills, with registered cases of pollution, etc.
Give the characteristics of the strains of microorganisms used in the work, where was it taken from and where was it deposited?
The information on the strain used is still not provided in sufficient form (was it previously described? By whom? Under what numbers are they deposited at?).
Lines 358-369 are not confirmed by visual results (chromatography, for example), a set of allegedly deleted components is presented.
The answers are presented, but most of them are with reference to Figure 1 (poorly readable/unreadable). Data on the mineralization of hydrocarbons have not been presented.
Figure 1 is not informative, low-quality. Figure 1c – what kind of component is left? It seems that it is not less than the sum of the peaks of the other components, which are shown in Fig. 1b.
Figure 1 is the fundamental key element for this article, and until it is reworked for easy perception, the article cannot be accepted.
Author Response
Point 1. 65: Error in Mn3 designation instead of Mn3+
Response 1. Already corrected in the line 65
Point 2. 78: Is PO4 the ionic form of a substance?
Response 2. Already corrected in the line 78
Point 3. 149: Dot after “Figure 3", correct to Figure 3 shows.
Response 3. Already corrected in the line 151
Point 4. 159: S. vulgare correct to S. Vulgare (in full italics).
Response 4. Already corrected in the line 161
Point 5. 253: In the phrase "In the figure 8 shows" correct English (this applies to the entire text. For example, there is a similar inaccuracy in the Discussion section).
Response 5. Already corrected in the line 254 and throughout the text.
Point 6. Detailed answers to questions have not been given (ignored) and comments that were made earlier have not been corrected (indicated in italics): What is the reason for choosing exactly these concentrations of oil for contamination and the duration of recultivation procedures with different components?
Response 6. We contaminated the soil with WMO at a discretionary concentration, as occurs when a soil is contaminated in a real way. While the duration of the experiment was conducted taking into account the results of previous experiments in which the effect of essential minerals and enzymes from different sources has been demonstrated.
Point 7. When working with soils with inducted pollution, concentrations already available in practice are taken as a reference. For example, at landfills, with registered cases of pollution, etc.
Response 7. When biological remediation methods are applied, it is not possible to start from the initial concentration of contamination. It is required to first remove the excess by mechanical removal or some other physical method, so that subsequent biological activity to remove the contaminant is possible.
Point 8 Give the characteristics of the strains of microorganisms used in the work, where was it taken from and where was it deposited?
Response 8. Rhizobium etli; that synthesizes phytohormones within the roots at the same time capable of mineralizing some aliphatic HCs, included aromatic by coometabolism [17-18]. Also, phytoremediation of soil polluted by WMO with S. vulgare could be enhanced with Rhizophagus irregularis, an endomycorrhizal fungus capable to solubilize P besides its tolerance to the aliphatic and aromatic phytotoxicity of WMO. R. etli and R. irregularis were taken from the microbial collection of the Environmental Microbiology Laboratory of the IIQB of the UMSNH. These strains were preserved in vials with sterile soil. This information is described in the lines: 74-79, 425-428.
Point 9. The information on the strain used is still not provided in sufficient form (was it previously described? By whom? Under what numbers are they deposited at?).
Response 9. Rhizobium etli; that synthesizes phytohormones within the roots at the same time capable of mineralizing some aliphatic HCs, included aromatic by coometabolism [17-18]. Also, phytoremediation of soil polluted by WMO with S. vulgare could be enhanced with Rhizophagus irregularis, an endomycorrhizal fungus capable to solubilize P besides its tolerance to the aliphatic and aromatic phytotoxicity of WMO. R. etli and R. irregularis were taken from the microbial collection of the Environmental Microbiology Laboratory of the IIQB of the UMSNH. These strains were preserved in vials with sterile soil. This information is described in the lines: 74-79, 425-428
Point 10. Lines 358-369 are not confirmed by visual results (chromatography, for example), a set of allegedly deleted components is presented.
Response 10. It has already been described with the results of the hydrocarbon identification table in the following lines 291- 295
Point 11. The answers are presented, but most of them are with reference to Figure 1 (poorly readable/unreadable). Data on the mineralization of hydrocarbons have not been presented.
Response 11. Figure 1 has already been improved on page 4. Also table 1 has already been added with the detection of hydrocarbons in the lines 291-297.
Point 12. Figure 1 is not informative, low-quality. Figure 1c – what kind of component is left? It seems that it is not less than the sum of the peaks of the other components, which are shown in Fig. 1b.
Response 12. Figure 1 has already been improved on page 4. Also table 1 has already been added with the detection of hydrocarbons in the lines 291-297.
Point 13. Figure 1 is the fundamental key element for this article, and until it is reworked for easy perception, the article cannot be accepted.
Response 13. Figure 1 has already been improved on page 4. Also table 1 has already been added with the detection of hydrocarbons in the lines 291-297.

Reviewer 4 Report
The text has been improved. I suggest just some corrections of stile to improve its clearness.
The commas referring to the concentrations in text and figures probably mean "thousands" and should be deleted, because in Europe this is read as commas. I recommend to give the concentration of waste motor oil used in the experiment, in percent to avoid unnecessary and non-measurable numbers.
Chapter "Material and methods" should be placed as the second. After the reader knows the experiment, the results can be understood in a better way.
Line 10: WMO 3,45 %, 6,54%, and 8,95 % (you cannot measure more exactly!) .... and all subsequent respective concentrations
Line 46: ... limit of HCs is 0,44 %
Line 83: ... contaminated by 3,45%, 6,54% and 8,98 % WMO by applied crude fungal extract ....
Line 92: ... biostimulated both with C. arietinum applied as GM, and CFE, decreased to 0,207%, ...
Line 118: The remaining peak given in fig. 1C is composed of ... (?); (the results of mass spectrometry are not mentioned here!)
figure 2: please add to "absolute control" no WMO
figure 2: please add to "relative control" plus 3,45% WMO
figure 2: 3,45 % WMO plus CFE at day 30
figure 2: 3,45 % WMO plus CFE at day 60
Lines 174-176: .... S. vulgare reached the highest AFW with 5,04 g in the soil polluted by WMO used as a relative control. In the soil not impacted by WMO, S. vulgare reached 4,36 g AFW irrigated only with water ....
figure 3: replace "radical length" by "root length"
Lines 198-203: ... applied CFE and GM. As relative control, S. vulgare was used in soil not polluted by WMO, and fed with 100% mineral solution, and uninocculated with R. irregularis and/or R. etli, which generated an ADW of 1,6 g followed by 1,08 of S. vulgare in soil not impacted by WMO irrigated only with water as an absolute control. In soil impacted by WMO, the ADW of S. vulgare inocculated with R. irregulare and/or R. etli reached an ADW of 0,7 g, while S. vulgare .,....
Line 223: ... 3,35 %
Line 225: ... 0,679 %
Line 227: ... 0,207%
Line 239-240: ...CFE and GM. The highest percentage of colonization in the root of S. vulgare yielded 79,2% ...
Line 242: ... control. This shows that ...
Line 270: Figure 1A shows ...
Line 276: ... populations. This was followed ...
Line 279: delete "while"
Line 373: 40 mesh = 0,42 mm, resp. 0,1682 mm = about 86 mesh! (Here it is usual to sieve < 2mm!)
Lines 427-428: in the abstract, you should mention that you had added H2O2, Cu and Mn !
Line 457: results from mass spectrometry are missing, figures 1 show the gas chromatogram. Add one sentence about the use of the mass spectra: amount of aromatics, hetero atoms O-N-S, reactive groups ?
Ref. 9) Marcela HRENIUC, Mirela COMAN, Bogdan CIORUŢA: CONSIDERATIONS REGARDING THE SOIL POLLUTION WITH OIL PRODUCTS IN SĂCEL - MARAMUREŞ, 20th Henry Coanda Air Force Academy Anniversary AFASES 2015, 557-562
- Ref. 32) Hussain I, Puschenreiter M., Soja G., Schöftner Ph., Sohail Y., Wange A. , Syed J.H., Reichenauer T.G.: Rhizoremediation of petroleum hydrocarbon-contaminated soils: Improvement opportunities and field applications. Environmental and Experimental Botany March 2018, 147, 202-219
Author Response
Point 1. The commas referring to the concentrations in text and figures probably mean "thousands" and should be deleted, because in Europe this is read as commas. I recommend to give the concentration of waste motor oil used in the experiment, in percent to avoid unnecessary and non-measurable numbers.
Response 1. Commas have already been removed
Point 3. Chapter "Material and methods" should be placed as the second. After the reader knows the experiment, the results can be understood in a better way.
Response 3. In the instructions for authors it is established that materials and methods should follow the discussion of results. Attached is the link where this information is corroborated. Plants | Instructions for Authors (mdpi.com)
Point 4. Line 10: WMO 3,45 %, 6,54%, and 8,95 % (you cannot measure more exactly!) .... and all subsequent respective concentrations
Response 4. We contaminate the soil in a discretionary manner, as occurs when soil is contaminated in real situations.
Point 5. Line 46: ... limit of HCs is 0,44 %
Response 5. Is 4400 ppm
Point 6. Line 83: ... contaminated by 3,45%, 6,54% and 8,98 % WMO by applied crude fungal extract ....
Response 6. Concentrations were left in parts per million without the commas.
Point 7. Line 92: ... biostimulated both with C. arietinum applied as GM, and CFE, decreased to 0,207%, ...
Response 7. Already corrected in the line 92
Point 8. Line 118: The remaining peak given in fig. 1C is composed of ... (?); (the results of mass spectrometry are not mentioned here!)
Response 8. Table 1 has already been added with the detection of hydrocarbons in the lines 291-297.
Point 9. figure 2: please add to "absolute control" no WMO
Response 9. The legends in figure 2 have been modified.
Point 10. figure 2: please add to "relative control" plus 3,45% WMO
Response 10. The legends in figure 2 have been modified
Point 11. figure 2: 3,45 % WMO plus CFE at day 30
Response 11. The legends in figure 2 have been modified
Point 12. figure 2: 3,45 % WMO plus CFE at day 60
Response 12. The legends in figure 2 have been modified
Point 13. Lines 174-176: .... S. vulgare reached the highest AFW with 5,04 g in the soil polluted by WMO used as a relative control. In the soil not impacted by WMO, S. vulgare reached 4,36 g AFW irrigated only with water ....
Response 13. It has been modified in lines 200-207
Point 14. figure 3: replace "radical length" by "root length"
Response 14. The legends in figure 3 have been modified
Point 15. Lines 198-203: ... applied CFE and GM. As relative control, S. vulgare was used in soil not polluted by WMO, and fed with 100% mineral solution, and uninocculated with R. irregularis and/or R. etli, which generated an ADW of 1,6 g followed by 1,08 of S. vulgare in soil not impacted by WMO irrigated only with water as an absolute control. In soil impacted by WMO, the ADW of S. vulgare inocculated with R. irregulare and/or R. etli reached an ADW of 0,7 g, while S. vulgare .,....
Response 15. It has been modified in lines 200-207
Point 16. Line 223: ... 3,35 %
Response 16. Concentrations were left in parts per million without the commas.
Point 17. Line 225: ... 0,679 %
Response 17. Concentrations were left in parts per million without the commas.
Point 18. Line 227: ... 0,207%
Response 18. Concentrations were left in parts per million without the commas.
Point 19. Line 239-240: ...CFE and GM. The highest percentage of colonization in the root of S. vulgare yielded 79,2% ...
Response 19. It has been modified in lines 241-242
Point 20. Line 242: ... control. This shows that ...
Response 20. It has been modified in line 244
Point 21. Line 270: Figure 1A shows ...
Response 21. It has been modified in line 298
Point 22. Line 276: ... populations. This was followed ...
Response 22. It has been modified in lines 304-305.
Point 23. Line 279: delete "while"
Response 23. The while has been removed
Point 24. Line 373: 40 mesh = 0,42 mm, resp. 0,1682 mm = about 86 mesh! (Here it is usual to sieve < 2mm!)
Response 24. It has been modified on line 417
Point 25. Lines 427-428: in the abstract, you should mention that you had added H2O2, Cu and Mn !
Response 25. The addition of these components was eliminated.
Point 26. Line 457: results from mass spectrometry are missing, figures 1 show the gas chromatogram. Add one sentence about the use of the mass spectra: amount of aromatics, hetero atoms O-N-S, reactive groups ?
Response 26. Table 1 has already been added with the detection of hydrocarbons in the lines 291-297.
Point 27. Ref. 9) Marcela HRENIUC, Mirela COMAN, Bogdan CIORUŢA: CONSIDERATIONS REGARDING THE SOIL POLLUTION WITH OIL PRODUCTS IN SĂCEL - MARAMUREŞ, 20th Henry Coanda Air Force Academy Anniversary AFASES 2015, 557-562
Response 27. The reference has been modified
Point 28. Ref. 32) Hussain I, Puschenreiter M., Soja G., Schöftner Ph., Sohail Y., Wange A. , Syed J.H., Reichenauer T.G.: Rhizoremediation of petroleum hydrocarbon-contaminated soils: Improvement opportunities and field applications. Environmental and Experimental Botany March 2018, 147, 202-219
Response 28. The reference has been modified

Round 3
Reviewer 3 Report
Only the introduction part has been improved. Regarding the methodological part and the presentation of the results, no qualitative improvement has been made. The table cannot replace high-quality chromatography, in extreme cases it can only be an addition. Figure 1 has not been fundamentally changed for the better, has not been improved. Answers to the questions raised that improve the quality of work were also not received. The article cannot be recommended for publication.Author Response
Response to reviewer 3
The observations made by the reviewers have been corrected and answered according to the following list, which can be corroborated directly in the manuscript.
Reviewers 1, 2 and 4 no longer sent comments after having answered their indications, so we assume that most of the reviewers agree with the publication of the manuscript.
Response to Reviewer 3 Round 2 Comments
Point 1. Error in Mn3 designation instead of Mn3+
Response 1. Already corrected in the line 65
Point 2. 78: Is PO4 the ionic form of a substance?
Response 2. Already corrected in the line 78
Point 3. 149: Dot after “Figure 3", correct to Figure 3 shows.
Response 3. Already corrected in the line 153
Point 4. 159: S. vulgare correct to S. Vulgare (in full italics).
Response 4. Already corrected in the line 161
Point 5. 253: In the phrase "In the figure 8 shows" correct English (this applies to the entire text. For example, there is a similar inaccuracy in the Discussion section).
Response 5. Already corrected in the line 256 and throughout the text.
Point 6. Detailed answers to questions have not been given (ignored) and comments that were made earlier have not been corrected (indicated in italics): What is the reason for choosing exactly these concentrations of oil for contamination and the duration of recultivation procedures with different components?
Response 6. The agricultural soil was contaminated in a discretionary way with the WMO because generally caused or accidental hydrocarbon spills happens like this. While the duration of the bioremediation experiment was based on previous results of biostimulation of soil impacted by WMO through the application of essential minerals to induce the aerobic heterotrophic aerobic microbiota to oxidize them, while the application of fungal extracellular enzyme extract that are stable to the conditions environmental conditions of the soil, which allows the hydrolysis and subsequent elimination of aromatic hydrocarbons
Point 7. When working with soils with inducted pollution, concentrations already available in practice are taken as a reference. For example, at landfills, with registered cases of pollution, etc.
Response 7. When agricultural soil is contaminated with mixtures of hydrocarbons such as WMO, it is impossible to initiate bioremediation with a biological action, since the high concentration of extremely toxic hydrocarbons prevents it. Therefore, the cleaning of the soil begins with a mechanical removal until a level of reduction of the WMO concentration that first allows the biostimulation that decreases the concentration to a level for sowing of plants that tolerate it and that can be more effective for the elimination. definition of the WMO as microorganisms that promote plant growth, the result is a concentration of hydrocarbons similar to or less than what is naturally detected in soil not contaminated with hydrocarbons for the reuse of soil for agricultural production that is safe for human consumption. and/or animal. For which the successful results of experiments carried out in soil with different concentrations of WMO, types of biostimulation and phytoremediation are used.
Point 8 Give the characteristics of the strains of microorganisms used in the work, where was it taken from and where was it deposited?
Response 8. Rhizobium etli; is an endophytic genus of plant growth-promoting bacteria isolated from Phaseolus vulgaris root nodules planted in soil with hydrocarbon contamination problems [17-18]. Rhizophagus irregularis is an endomycorrhizal fungus isolated from grasses grown in oil-contaminated soils. R. etli was identified according to methods accepted by the Bargery´s Manual, while R. irregularis was identified based on a molecular profile. Both strains were taken from the microbial collection of the Environmental Microbiology Laboratory of the Institute of Biological Chemical Research of the UMSNH. These strains were maintained in vials with sterile agricultural soil. This information is indicated in the lines: 74-81, 470-473 of this article.
Point 9. The information on the strain used is still not provided in sufficient form (was it previously described? By whom? Under what numbers are they deposited at?).
Response 9. Rhizobium etli strain has been used for research on biostimulation of agricultural soil impacted by WMO p of our authorship written in Spanish previously published in Journal of the Selva Andina Research Society. This strain has biochemical and tolerance to hydrocarbons characteristics that are common to others in international collections, which is why they have never been deposited anywhere. In the same way, the strain of Rhizophagus irregularis has been used in master's studies in our laboratory, it has physiological characteristics common to strains recovered from roots of domestic crops such as beans, similar to what is reported in the literature, and they have not been deposited either. in any collection given the commonness of its behavior in plants.
Point 10. Lines 358-369 are not confirmed by visual results (chromatography, for example), a set of allegedly deleted components is presented.
Response 10. It has already been described with the results of the hydrocarbon identification table of remnant hydrocarbons derived from mass coupled gas chromatography analysis of soil with 34500 ppm of waste motor oil before and after biostimulation, in the following lines 295-296
Point 11. The answers are presented, but most of them are with reference to Figure 1 (poorly readable/unreadable). Data on the mineralization of hydrocarbons have not been presented.
Response 11. Figure 1 has already been improved on page 4. Also, table 1 has already been added with remnant hydrocarbons derived from mass coupled gas chromatography analysis in the lines 295-296
Point 12. Figure 1 is not informative, low-quality. Figure 1c – what kind of component is left? It seems that it is not less than the sum of the peaks of the other components, which are shown in Fig. 1b.
Response 12. Figure 1 has already been improved on page 4. Also, table 1 has already been added with remnant hydrocarbons derived from mass coupled gas chromatography analysis in the lines 295-296
Point 13. Figure 1 is the fundamental key element for this article, and until it is reworked for easy perception, the article cannot be accepted.
Response 13. Figure 1 has already been improved on page 4. Also, table 1 has already been added with remnant hydrocarbons derived from mass coupled gas chromatography analysis in the lines 295-296.
Response to Reviewer 4 Round 2 Comments
Point 1. The commas referring to the concentrations in text and figures probably mean "thousands" and should be deleted, because in Europe this is read as commas. I recommend to give the concentration of waste motor oil used in the of biostimulation and phytoremediation experiment, in percent to avoid unnecessary and non-measurable numbers.
Response 1. Commas have already been removed
Point 3. Chapter "Material and methods" should be placed as the second. After the reader knows the experiment, the results can be understood in a better way.
Response 3. In the instructions for authors it is established that materials and methods should follow the discussion of results. Attached is the link where this information is corroborated. Plants | Instructions for Authors (mdpi.com)
Point 4. Line 10: WMO 3,45 %, 6,54%, and 8,95 % (you cannot measure more exactly!) .... and all subsequent respective concentrations
Response 4. The agricultural soil was contaminated in a discretionary way with the WMO because generally caused or accidental hydrocarbon spills happens like this. While the duration of the bioremediation experiment was based on previous results of biostimulation of soil impacted by WMO through the application of essential minerals to induce the aerobic heterotrophic aerobic microbiota to oxidize them, while the application of fungal extracellular enzyme extract that are stable to the conditions environmental conditions of the soil, which allows the hydrolysis and subsequent elimination of aromatic hydrocarbons
Point 5. Line 46: ... limit of HCs is 0,44 %
Response 5. Is 4400 ppm
Point 6. Line 83: ... contaminated by 3,45%, 6,54% and 8,98 % WMO by applied crude fungal extract ....
Response 6. Concentrations were left in parts per million without the commas.
Point 7. Line 92: ... biostimulated both with C. arietinum applied as green manure (GM), and crude fungal extract (CFE), decreased to 0,207%, ...
Response 7. Already corrected in the line 93
Point 8. Line 118: The remaining peak given in fig. 1C is composed of ... (?); (the results of mass spectrometry are not mentioned here!)
Response 8. table 1 has already been added with remnant hydrocarbons derived from mass coupled gas chromatography analysis in the lines 295-296
Point 9. figure 2: please add to "absolute control" no WMO
Response 9. The legends in figure 2 have been modified.
Point 10. figure 2: please add to "relative control" plus 3,45% WMO
Response 10. The legends in figure 2 have been modified
Point 11. figure 2: 3,45 % WMO plus CFE at day 30
Response 11. The legends in figure 2 have been modified
Point 12. figure 2: 3,45 % WMO plus CFE at day 60
Response 12. The legends in figure 2 have been modified
Point 13. Lines 174-176: .... S. vulgare reached the highest AFW with 5,04 g in the soil polluted by WMO used as a relative control. In the soil not impacted by WMO, S. vulgare reached 4,36 g AFW irrigated only with water ....
Response 13. It has been modified in lines 178-180
Point 14. figure 3: replace "radical length" by "root length"
Response 14. The legends in figure 3 have been modified
Point 15. Lines 198-203: ... applied CFE and GM. As relative control, S. vulgare was used in soil not polluted by WMO, and fed with 100% mineral solution, and uninoculated with R. irregularis and/or R. etli, which generated an ADW of 1,6 g followed by 1,08 of S. vulgare in soil not impacted by WMO irrigated only with water as an absolute control. In soil impacted by WMO, the ADW of S. vulgare inoculated with R. irregularis and/or R. etli reached an ADW of 0,7 g, while S. vulgare .,....
Response 15. It has been modified in lines 200-207
Point 16. Line 223: ... 3,35 %
Response 16. Concentrations were left in parts per million without the commas.
Point 17. Line 225: ... 0,679 %
Response 17. Concentrations were left in parts per million without the commas.
Point 18. Line 227: ... 0,207%
Response 18. Concentrations were left in parts per million without the commas.
Point 19. Line 239-240: ...CFE and GM. The highest percentage of colonization in the root of S. vulgare yielded 79,2% ...
Response 19. It has been modified in lines 243-244
Point 20. Line 242: ... control. This shows that ...
Response 20. It has been modified in line 246
Point 21. Line 270: Figure 1A shows ...
Response 21. It has been modified in line 299
Point 22. Line 276: ... populations. This was followed ...
Response 22. It has been modified in lines 305-306.
Point 23. Line 279: delete "while"
Response 23. The while has been removed
Point 24. Line 373: 40 mesh = 0,42 mm, resp. 0,1682 mm = about 86 mesh! (Here it is usual to sieve < 2mm!)
Response 24. It has been modified on line 418
Point 25. Lines 427-428: in the abstract, you should mention that you had added H2O2, Cu and Mn !
Response 25. The addition of these components was eliminated.
Point 26. Line 457: results from mass spectrometry are missing, figures 1 show the gas chromatogram. Add one sentence about the use of the mass spectra: amount of aromatics, hetero atoms O-N-S, reactive groups ?
Response 26. Table 1 has already been added with remnant hydrocarbons derived from mass coupled gas chromatography analysis in the lines 295-296.
Point 27. Ref. 9) Marcela Hreniuc, Mirela Coman, Bogdan Cioruţa: Considerations regarding the soil pollution with oil products in Săcel - Maramureş, 20th Henry Coanda Air Force Academy Anniversary AFASES 2015, 557-562
Response 27. The reference has been modified
Point 28. Ref. 32) Hussain I, Puschenreiter M., Soja G., Schöftner Ph., Sohail Y., Wange A. , Syed J.H., Reichenauer T.G.: Rhizoremediation of petroleum hydrocarbon-contaminated soils: Improvement opportunities and field applications. Environmental and Experimental Botany March 2018, 147, 202-219
Response 28. The reference has been modified
